Extracellular vesicles in patients in the acute phase of psychosis and after clinical improvement: an explorative study

Tunset Mette Elise mette.elise.tunset@stolav.no 1 2
Haslene-Hox Hanne 3
Van Den Bossche Tim 4 5
Vaaler Arne Einar 1 2
Sulheim Einar 3 6
Kondziella Daniel 7 8
1 Department of Østmarka- Division of Mental Healthcare, St. Olavs University Hospital , Trondheim , Norway
2 Department of Mental Health- Faculty of Medicine and Health Sciences, Norwegian University of Science and Technology (NTNU) , Trondheim , Norway
3 Department of Biotechnology and Nanomedicine, SINTEF , Trondheim , Norway
4 VIB - UGent Center for Medical Biotechnology, VIB , Ghent , Belgium
5 Department of Biomolecular Medicine, Faculty of Medicine and Health Sciences, Ghent University , Ghent , Belgium
6 Department of Physics, Norwegian University of Science and Technology (NTNU) , Trondheim , Norway
7 Department of Neurology, Rigshospitalet, Copenhagen University Hospital , Copenhagen , Denmark
8 Department of Clinical Medicine, Faculty of Health and Medical Sciences, University of Copenhagen , Copenhagen , Denmark
Gollo Leonardo
Electronic publication date: 2020 Aug 26
Publication date: 2020
Volume: 8
Electronic Location ID: e9714
Received 2020 May 5; Accepted 2020 Jul 23
Copyright: ©2020 Tunset et al.
Copyright year: 2020
Copyright holder: Tunset et al.
License: This is an open access article distributed under the terms of the Creative Commons Attribution License, which permits unrestricted use, distribution, reproduction and adaptation in any medium and for any purpose provided that it is properly attributed. For attribution, the original author(s), title, publication source (PeerJ) and either DOI or URL of the article must be cited.
License URL: https://creativecommons.org/licenses/by/4.0/

Keywords: Psychosis, Proteomics, Extracellular vesicles, Gene ontology, Blood brain barrier, Glymphatics, Substance abuse, Immune system, Glutamatergic synapses, Brain enriched proteins

Funding: St. Olavs University Hospital ref16/9564-97/NISLIN ref17/10533-124/NISLIN Research Foundation—Flanders 1S90918N This work was supported by St. Olavs University Hospital (ref16/9564-97/NISLIN and ref17/10533-124/NISLIN) and by the Research Foundation—Flanders through an FWO SB PhD Grant (1S90918N). The funders had no role in study design, data collection and analysis, decision to publish, or preparation of the manuscript.

==============================
Extracellular vesicles (EVs) are cell-derived structures that transport proteins, lipids and nucleic acids between cells, thereby affecting the phenotype of the recipient cell. As the content of EVs reflects the status of the originating cell, EVs can have potential as biomarkers. Identifying EVs, including their cells of origin and their cargo, may provide insights in the pathophysiology of psychosis. Here, we present an in-depth analysis and proteomics of EVs from peripheral blood in patients (n = 25) during and after the acute phase of psychosis. Concentration and protein content of EVs in psychotic patients were twofold higher than in 25 age- and sex-matched healthy controls (p < 0.001 for both concentration and protein content), and the diameter of EVs was larger in patients (p = 0.02). Properties of EVs did not differ significantly in blood sampled during and after the acute psychotic episode. Proteomic analyses on isolated EVs from individual patients revealed 1,853 proteins, whereof 45 were brain-elevated proteins. Of these, five proteins involved in regulation of plasticity of glutamatergic synapses were significantly different in psychotic patients compared to controls; neurogranin (NRGN), neuron-specific calcium-binding protein hippocalcin (HPCA), kalirin (KALRN), beta-adducin (ADD2) and ankyrin-2 (ANK2). To summarize, our results show that peripheral EVs in psychotic patients are different from those in healthy controls and point at alterations on the glutamatergic system. We suggest that EVs allow investigation of blood-borne brain-originating biological material and that their role as biomarkers in patients with psychotic disorders is worthy of further exploration.

Introduction

Extracellular vesicles (EVs) are nanoscale (30–1,000 nm), cell-derived, double-lipid membrane structures containing proteins, RNAs and lipids (Van Niel, D’Angelo & Raposo, 2018; Yanez-Mo et al., 2015). They are secreted from cells by direct budding of the cell membrane (microvesicles) or by exocytosis of multivesicular bodies (exosomes). EVs are involved in signaling between cells and their cargo is not random but controlled by the originating cells (Van Niel, D’Angelo & Raposo, 2018; Yanez-Mo et al., 2015). The proteins in EVs are common to the cells of origin, which allows for identificaion of EV origin by analyzing cell-specific proteins (Van Niel, D’Angelo & Raposo, 2018; Yanez-Mo et al., 2015).

Evidence suggests that EVs from the brain are present in peripheral blood (Galbo Jr et al., 2017; Goetzl et al., 2016a; Kapogiannis et al., 2019b). Hence, sampling of blood-borne EVs may be a non-invasive way to gain access to brain-derived biological material. Since evidence indicates that EVs are involved in brain plasticity and information storage (Chivet et al., 2014; Fowler, 2019; Goldie et al., 2014; Pastuzyn et al., 2018; Yanez-Mo et al., 2015), research on EVs may reveal novel insights into brain disorders in which these processes are relevant, including psychosis. Further, psychotic disorders are associated with abnormalities in several organ systems other than the brain (Pillinger et al., 2018), supporting the rational of investigating peripheral EVs in patients with psychosis. Thus, identifying EVs, their cells of origin and their cargo may uncover insights in the pathophysiology of psychosis and may serve as a source for biomarkers. In theory, EVs could also be used as therapeutic vehicles, as evidence indicates that their membrane proteins can guide them to specific recipient cells (Van Niel, D’Angelo & Raposo, 2018; Yanez-Mo et al., 2015). To our knowledge, there are only four published studies on psychosis and EVs: one based on brain biopsies (Banigan et al., 2013), a case report involving cerebrospinal fluid (CSF) analysis (Mobarrez et al., 2013), and two studies showing altered insulin signaling in L1 cell adhesion molecule (L1CAM) positive EVs in patients with schizophrenia (Kapogiannis et al., 2019a; Wijtenburg et al., 2019).

In the present study, we investigated if concentration, size and protein content of EVs differed between psychotic patients and controls, and if the state of the disease affected these characteristics. We also explored whether brain-derived EVs could be detected in peripheral blood, and if the pattern of brain-elevated proteins was different in patients compared to controls. Finally, we used gene ontology (GO) analysis of the proteome to explore which protein categories were over-represented in significantly changed proteins.

Materials and Methods

Study participants

A total of 25 psychotic patients (six females, mean age 33.1 ± 11.0 years), during a first episode of psychosis or during acute exacerbation of a known psychotic disorder, were recruited between December 2016 to December 2018 from the Østmarka acute inpatient psychiatric department, St. Olavs University Hospital, Trondheim, Norway. Exclusion criteria were affective psychoses, heart diseases, neurological diseases, pregnancy, rheumatic diseases, autoimmune diseases and cancer. In addition, patients with organic causes of psychosis were excluded. Diagnosis was assessed by ICD 10 Criteria for research and registered after discharge from hospital. Among the 25 patients, 12 (48%) had schizophrenia, 4 (16%) substance-induced psychotic disorder, 3 (12%) acute psychosis, 3 (12%) had unspecified psychosis and 3 (12%) had other psychotic disorders. Mean time since onset of the first psychotic episode was 63 months ± 81 months (if no earlier episode, time since symptom debut of the present episode was registered).

A first blood sample was taken at inclusion during the acute phase of psychosis (T1). A second blood sample was drawn 6 weeks later or more (T2), when patients were clinically back to baseline or much/very much improved according to the Clinical Global Impression- Improvement Scale (CGI-I). The time between sample time points was 79  ± 34 days (range: 42–162 days). The second blood sample was collected from 18 patients. Seven patients were lost to follow-up. Ongoing abuse of recreational drugs was screened for by questioning and a urine drug screen at first sampling time point and by questioning at second sampling point.

Healthy control persons (n = 25) were recruited among the staff of the Department of Psychiatry, Østmarka, St. Olavs University Hospital, Trondheim. Controls were matched to psychotic patients according to sex and age (+/−5 years). Mean age of healthy controls was 34.2 ± 11.2 years. Exclusion criteria were the same as for the psychotic patient, including (self-reported) illegal substance use.

Scoring range

Clinical Global Impression-Severity Scale (CGI-S) scores were registered at both sampling points by the psychologist, board-certified psychiatrist or psychiatric resident in charge of the patient. The CGI-S ranges from 1 (“normal”) to 7 (“among the most extremely ill”) (Guy, 1976).

Blood sampling and EV isolation

Blood (15 ml) was collected by venipuncture in patients at the two sampling points, and in control persons, with EDTA as anti-coagulant. The samples were kept on ice and centrifuged fresh (2,000 g, 30 min, 4 °C, Hettich Rotina 420R centrifuge with rotor number 4723) within 2 h to isolate cell free plasma. Plasma (6 ml) was transferred to Eppendorf tubes and centrifuged (10,000 g, 30 min, 4 °C , Eppendorf 5418R centrifuge with rotor number FA-45-18-11). Supernatant was transferred to cryotubes, and both pellets and supernatants were frozen at −80 °C awaiting further analysis. All samples were further processed within 1 year.

Pellet fractions were thawed in room temperature and resuspended in 100 µl phosphate-buffered saline (PBS), and pellet samples originating from the same blood sample were pooled. The samples were centrifuged again to remove any residual cells and debris, first at 2,000 g (30 min, 4 °C, Eppendorf 5417R with rotor number FA-45-30-11). The supernatant was transferred to a pre-weighed Eppendorf tube and centrifuged at 10,000 g (30 min, 4 °C). The resulting supernatant was discarded, and the pellet was resuspended in ammonium bicarbonate buffer (100 µl, 100 mM) for further analysis. Samples for proteomics were frozen at −80 °C in Protein LoBind Eppendorf tubes before further sample processing (approximately 1 month). The concentration and separation approach aimed to provide an EV sample in the high recovery, low specificity category of MISEV2018 guidelines (Thery et al., 2018). We have submitted all relevant data of our experiments to the EV-TRACK knowledgebase (EV-TRACK ID: EV200067) (Van Deun et al., 2017).

Characterization of isolated EV samples

The protein concentration in isolated EV samples was determined by Qubit Quant-IT Protein Assay Kit (Thermo Fisher Scientific, cat. no. Q33211) on a Qubit Fluorometer 2.0. EVs were analyzed for size and concentration using Nanoparticle Tracking Analysis (NTA, Nanosight LM10, Malvern Panalytical Ltd, Malvern, UK). EVs were diluted 100-fold in sterile PBS and three one-minute movies were recorded on the NTA (detection threshold 4, auto blur size, max jump distance).

Statistical analyses

A two-sample (un-paired) t-test was used to compare the mean values of main characteristics of EVs (size, concentration and protein content) between the patients in the acute phase of psychosis and healthy controls, whereas a paired sample t-test was used to examine if main characteristics of EV changed from the acute psychotic phase (T1) to improvement of the psychotic episode (T2). To assess if a longer history of psychosis affected main characteristics of EVs, we compared patients with <1 year since onset of first psychosis with patients with 1 year or more since onset of first psychosis using two-sample t-test (un-paired). A two-sample t-test was also used to assess if drug abuse at the acute phase changed the characteristics of EVs.

Proteomics of isolated EVs

The protein composition of EVs was determined by LC-MS/MS analysis (Choi et al., 2015). Sample containing 30 µg of protein as determined by Qubit was diluted to 25 µL in ammonium bicarbonate buffer (100 mM), digested by trypsin and desalted as described earlier (Haslene-Hox et al., 2011). The sample was loaded and desalted on a pre-column for 5 min (Acclaim PepMap 100, 2 cm × 75 µm ID nanoViper column, packed with 3 µm C18 beads, flow rate: 5 µl/min, mobile phase: 0.1% TFA). For peptide separation a biphasic ACN gradient from two nanoflow UPLC pumps was used (flow rate of 250 nl/min, 120 min run) on a 25 cm analytical column (PepMap RSLC, 25 cm × 75 µm ID EASY-spray column, packed with µm C18 beads with pore size 100 Å). Solvent A was 0.1% FA (vol/vol) in water and solvent B was 100% ACN. The gradient composition was 5%B during trapping (5 min), 5–7%B (0.5 min), 7–22%B (59.5 min), 22–35%B (22 min), and 35–90%B (5 min). To wash the column between samples, elution of very hydrophobic peptides and conditioning of the column were performed during 10 min isocratic elution with 80%B and 15 min isocratic elution with 5%B. Peptides were ionized in the electrospray and analyzed by an Orbitrap Q-Exactive HF. Data-dependent-acquisition (DDA) mode was used to automatically switch between full scan MS and MS/MS acquisition. Q-Exactive HF Tune 2.9 was used for instrument control, and XCalibur 4.1 Survey full scan MS spectra (m/z 375-1500) were acquired with resolution R = 120,000 at m/z 200, automatic gain control (AGC) target of 3e6 and a maximum injection time of 100 ms. The 12 most intense eluting peptides above an intensity threshold of 50,000 counts, and charge states 2 to 5, were sequentially isolated to a target value (AGC) of 1e5 and a maximum injection time of 110 ms in the C-trap. Isolation width was maintained at 1.6 m/z (offset of 0.3 m/z), before fragmentation in the HCD (Higher-Energy Collision Dissociation) cell. The minimum AGC target for fragmentation were set at 5.5e3. Normalized collision energy (NCE) was 28% at fragmentation. Fragments were detected at a resolution of 60,000 at m/z 200, and first mass was fixed at m/z 120. One MS/MS spectrum of a precursor mass was allowed before dynamic exclusion for 20s with “exclude isotopes” on. Lock-mass internal calibration (m/z 445.12003) was enabled. Ion spray voltage was 1,800 V, no sheath and auxiliary gas flow, and capillary temperature was 275 °C.

In total, 68 samples were submitted to proteomic analysis, and each sample was analysed a single time with mass spectrometry in random order. The samples were divided in three groups: patient samples at first time point (n = 25); patient samples at second time point (n = 18); and control samples from age-matched healthy persons (n = 25).

Database search parameter and acceptance criteria for identification

The raw data was converted to Mascot Generic Format (mgf) peak lists with MS convert with peak picking of MS2 to convert to centroid data (Chambers et al., 2012). Peak lists obtained from MS/MS spectra were identified using X!Tandem (X!Tandem Vengeance,  v2015.12.15.2). The search was conducted using SearchGUI (v3.3.15). Protein identification was conducted against a concatenated target/decoy database of Homo sapiens (reference proteome downloaded from UniProtKB in March 2018) (Apweiler et al., 2004) with porcine trypsin (P00761) added as possible contaminant (40,660 entries in concatenated database, based on 20,330 entries from uniprot.org). A reverse target sequence decoy database was created in SearchGUI (Barsnes & Vaudel, 2018). The identification was done with specific trypsin digest and maximum two missed cleavages. Tolerance was set to 10 ppm for MS1 and 0.02 Da for MS2. Fixed modifications was Carbamidomethylation of C (+57.021464 Da). Variable modification was Oxidation of M (+15.994915 Da). In addition modifications during refinement procedure were used: Carbamidomethylation of C (+57.021464 Da, fixed), Acetylation of protein N-term (+42.010565 Da, variable), Pyrolidone from E (−18.010565 Da, variable), Pyrolidone from Q (−17.026549 Da, variable), Pyrolidone from carbamidomethylated C (−17.026549 Da, variable).

PeptideShaker (v1.16.38) (Vaudel et al., 2015) was used to infer peptides and proteins from SearchGUI spectrum identification results. Peptide Spectrum Matches (PSMs), peptides and proteins were validated at a 1.0% False Discovery Rate (FDR) estimated using the decoy hit distribution. Post-translational modification localizations were scored using the D-score (Vaudel et al., 2013). All samples were processed in parallel in PeptideShaker to provide data for all identified proteins across all samples, with individual quantitative measures for each sample. The average precursor intensity, an average of MS1 signal for all spectra allocated to a given protein in a given sample, was used for label-free quantitative evaluation.

Proteomic data analysis and submission of data to a repository

For quantification, average precursor intensities were normalized by dividing the intensity on the sum of intensities within individual samples. Statistical analysis was performed using Perseus (version 1.6.5.0) (Tyanova et al., 2016). Identification of significant differences in protein detection between sample groups were analysed in Perseus, using Student’s t-test with correction for multiple hypothesis testing by using permutation-based FDR < 0.01 and artificial within group variance s0 = 0.1. Missing values were imputed from a normal distribution with a 1.8 standard deviation shift from the average and a width of 0.3.

Gene ontology of identified proteins was analysed by PANTHER classification system (version 15.0, released on 2020-04-07) (Mi, Muruganujan & Thomas, 2013). The EV proteome was screened for brain-enriched proteins and membrane-proteins as determined in the human protein atlas (The Human protein Atlas, 2018; Uhlen et al., 2015). To control for co-isolation of lipoprotein and chylomicrons, we searched proteomic results for apolipoproteins (Karimi et al., 2018). The mass spectrometry data along with the identification results have been deposited to the ProteomeXchange Consortium (Vizcaino et al., 2014) via the PRIDE partner repository (Vizcaino et al., 2016) available at http://proteomecentral.proteomexchange.org/cgi/GetDataset with the dataset identifier PXD016293. EV proteome will also be deposited in the Vesiclepedia (Kalra et al., 2012).

Ethics

The study was approved by the Regional Ethics committee, South East Norway (2016/949). All participants gave their written, informed consent after a board-certified psychiatrist or psychologist had checked that they were able to do so.

Results

Clinical global impression-severity scale

The CGI-S score was used to evaluate the severeness of psychosis, and the change in state for patients in the two sample time points. The CGI-S score showed a decline from a median of 7 (defined as “among the most extremely ill patients”), range 5 to 7, during the acute psychotic period (T1) to a median of 4 (“moderately ill”), range 2-6, at T2. All patients had a lower CGI-S score at the second time point.

Size, concentration and protein content

Mean size and concentration of EVs and protein concentration in EV fractions are shown in Fig. 1. The protein concentration per EV was equal for all groups, and averaged at 1.2 ⋅10−6 ± 6.7 ⋅10−7 µg protein/vesicles. NTA analysis showed that most vesicles in the samples was between 75 and 200 nm in size, but larger vesicles were also present. Exosomes are defined as vesicles 30–150 nm in size (Yanez-Mo et al., 2015), and the isolated samples are likely a mixture of exosomes and microvesicles(the total population of vesicles is further referred to as EVs). We found that the size, concentration and protein content of EVs from psychotic patients differed significantly from healthy controls (Table 1A). There was no apparent difference between T1 and T2 in the psychotic patients (Table 1B). There were no significant differences either in EV characteristics between patients with a short (<1 year) versus longer (≥ 1 year) history of psychosis (Table 1C).

Figure 1 Concentration, size and protein content of EVs.

Vesicle concentration (A), vesicle concentration versus size from nano-tracking analysis (B), vesicle diameter (C) and protein concentration (D) of isolated EV fractions for psychotic patients during psychosis (T1) and in improved state (T2) and healthy controls (HC). P-values are given for significant differences. Bar shows mean value with standard deviation error bars.

Table 1 Main characteristics of EVs.

(A) Main characteristics of EVs in patients at first sampling point (T1) and controls. Values given as mean (SD). (B) Main characteristics in EVs from the 18 patients with complete data at both sampling time points (T1 and T2). Values given as mean (SD). (C) Main characteristics of EVs according to time since debut of psychosis at T1. Values given as mean (SD).

Table 1A: T1 vs HC	Psychotic patients (T1, n = 25)	Healthy Controls (HC, n = 25)	p-valuea	
Diameter of EVs (nm)	195 (20)	180 (12)	0.002b	
Concentration of EVs (particles/ml plasma)	2.4x107 (1,1x107)	1.2x107 (5.0x106)	<0.001b	
Protein content EVs (µg/ml plasma)	28.2(15.4)	13.3(8.9)	<0.001b	
Table 1B: T1 vs T2	Acute phase (T1)	Improved (T2)	Change	95% CI	p-valuec	
CGI score, median	7	4				
CGI score, mean	6.5 (0.65)	3.8 (1.23)				
Diameter of EVs (nm)	199 (18)	200 (25)	1	−17–13	0.798	
Concentration of EVs (particles/ml plasma)	2.2 × 107 (1.0 × 107)	2.3 × 107 (1.3 × 107)	9.5 × 105	−8.6 × 106–6.7 × 106	0.796	
Protein concentration in EV fraction (µg/ml plasma)	26.8(12.0)	21.4(13.1)	−5.4	−5.1 –15.9	0.294	
Table 1C: T1	Psychotic patients (T1, n = 25)	Mean	95% CI interval	p-valuea	
Years since debut of psychosis	<1 (n = 10)	≥ 1 (n = 15)				
Months since debut of first psychosis	4.1(4.8)	102.4(83.7)				
Concentration of EVs (particles/ml plasma)	2.0 × 107 (8.3 × 106)	2.7x107(1.2 × 107)	−7.4 × 106	−1.6 × 107–1.7 × 106	0.106	
Diameter of EVs (nm)	193(14)	196(23)	−3	−20–13	0.678	
Protein concentration in EV fraction (µg/ml plasma)	22.4(8.5)	32.0(17.9)	9.6	−22.2–3.0	0.130	
Notes.

a Two sample t-test.

b Equal variance not assumed.

c Paired t-test.

Substance use

About 40% of patients with schizophrenia spectrum disorders also have a substance use disorder (Hunt et al., 2018). Substance use disorders are highly correlated to smoking (Smith, Mazure & McKee, 2014) and linked with poor outcomes in symptom severity and service use in patients with psychosis (Abdel-Baki et al., 2017). We studied if a recent intake of illegal substances affected EV parameters. Nine and 3 patients had used illegal drugs within 1 week before T1 and T2, respectively. Within the psychosis group there was no significant change in size of EVs in the group without illegal substance use the week before sampling (191 nm) compared to patients with illegal substance use 1 week before sampling (203 nm) (n = 25, mean change 12 nm, p = 0.156, equal variance not assumed) at T1. There were no differences comparing mean concentrations in patients without (2.46 × 107 particles/ml) and with illegal substance use 1 week before sampling (2.19 × 107 particles/ml) at T1 (mean change 2.75 × 106 particles/ml n = 25, p = 0.573). There was no significant change in mean protein content in EV fractions in patients without (28.54 µg/ml) and with illegal substance use 1 week before sampling (2.765 µg/ml) (n = 25, mean change 0.89 µg/ml, p = 0.897).

EV proteomes

The protein cargo of EVs are central to understand their origin, function and classification (Choi et al., 2015). Shotgun proteomics of all EV samples resulted in 1,853 identified proteins with more than 1 identified peptide across all samples by 26,537 unique peptides, using a false positive rate (FDR) of 1% (Table S1). Of these, 1,658 (89%) proteins were identified in all three sample groups (Hulsen, De Vlieg & Alkema, 2008). 118 proteins were identified at one or two time points from the psychotic patient samples, while not detected in the control group (Fig. 2A). To verify the EV origin, the proteomes were compared to the 100 most frequent proteins found in exosomes from Exocarta (2018). We identified 93 of these in our sample material, without difference between patient samples and controls. This includes known positive EV markers including cytosolic proteins recovered in EVs (Alix (Q8WUM4), Tsg101 (Q99816)) and transmembrane or GPI-anchored proteins associated to plasma membranes or endosomes (CD9 (P21926), CD81 (P60033) and CD63 (P08962)), demonstrating that the isolated EV fractions in this study contained EVs (Thery et al., 2018). The enrichment of the EV proteomes was also confirmed by comparing the 1853 proteins identified with the human proteome by GO enrichment analysis. The GO-terms extracellular exosomes and vesicle-mediated transport were among the mostly enriched GO-terms (Table S2A), verifying that the EV isolation process yielded an EV-enriched fraction.

Figure 2 Overall proteomic findings.

(A) Venn diagram showing total number of identified proteins in psychotic patients during psychosis (T1) and in improved state (T2) and healthy controls (HC) and the overlap between sample groups. (B) Volcano plot showing the p-value versus the fold change of all proteins for all three groups compared pairwise with each other (Student’s t-test with multiple hypothesis correction, lines showing significance threshold (Significant at p < 0.05, Artificial within groups variance s0 = 0.1)).

Differentially expressed proteins were identified by comparing proteins in groups pairwise by normalized average precursor intensity of identified proteins (Fig. 2B). No proteins were identified as differentially expressed between T1 and T2, although COP9 signalosome complex subunit 6 (Q7L5N1) had significant p-value (p < 0.001) and close to significant fold change (0.66). In T1 and T2, 119 and 40 proteins were differentially expressed compared with healthy controls (HC), respectively (complete lists in Table S3). Combined, 131 proteins were differently expressed in T1 and/or T2 compared with HC, 102 proteins had increased abundance in T1 and/or T2 compared with HC, while 29 proteins had a lower abundance. The relative variance and distribution for each protein between samples within one group was considered to evaluate different heterogeneity in the three sample groups, and were similar for HC, T1 and T2 groups. Thus, the group heterogeneity was similar for HC, T1 and T2.

The proteins that were differentially expressed in psychotic patients compared with healthy controls were submitted to GO overrepresentation analysis, to identify enriched GO terms for the changed proteins (significantly enriched GO terms compared to the human proteome, FDR threshold at 0.05) (Tables S2B and S2C).

Proteins with higher abundance in psychotic patients had overrepresented GO-terms related to localization and transport inside and out of the cell, as well as leukocyte and neutrophil activation. GO terms enriched for proteins that had a higher abundance in HC samples were represented by lipoprotein processes, the immunoglobulin complex and complement pathway. Of note, the GO terms main axon and postsynapse were enriched in the proteins with higher abundance in healthy controls and represented 9 proteins (Table S4).

The mass spectrometry data along with the identification results have been deposited to the ProteomeXchange Consortium (Vizcaino et al., 2014) via the PRIDE partner repository (Vizcaino et al., 2016) available at http://proteomecentral.proteomexchange.org/cgi/GetDataset with the dataset identifier PXD016293 and Project DOI 10.6019/PXD016293.

Lipoproteins

Lipoproteins are an important constituent of EVs and occur also in plasma as lipid particles that can co-isolate with EVs (Raposo & Stoorvogel, 2013). Sixteen apolipoproteins were identified by proteomics, including suggested markers for non-EV co-isolated structures (Apolipoprotein A1/2 and B). Significant differences with higher levels in healthy controls were found for Apolipoprotein L1, B-100, A-I and A-IV (Table S5). The apolipoproteins contributed with 1.0 to 6.0% of the total signal intensity for each sample analysed by proteomics (average 2.3%), indicating that the overall contribution of lipoproteins in the samples are low. The percentage contribution of apolipoprotein spectra was higher in healthy controls (2.6 ± 1.0) compared with T1 (2.0 ± 0.9) (p = 0.04, unpaired t-test), but not T2. The GO term for chylomicron and lipid-particle formation was enriched in proteins more abundant in healthy controls, also corroborating that the concentration of lipid particles in HC compared with psychotic patients is proportionally higher.

Brain proteins

The overall EV proteome is not dominated by brain elevated proteins, but include proteins from EVs originating from blood cells, immune cells and endothelial cells in addition to tissue-derived EVs. Proteins that can originate from the brain was identified by comparison of our EV proteome with the list of genes with an elevated expression in the brain compared to other tissue types in the tissue atlas of Human Protein Atlas (The Human Protein Atlas, 2018; Uhlen et al., 2015). These proteins have at least a four-fold higher mRNA level in the brain compared to the average level in all other tissues according to The Human Protein Atlas. We identified 45 proteins in our EV proteome (Table S6) that were also found as elevated in brain in the Human Protein Atlas (The Human protein Atlas, 2018; Uhlen et al., 2015). The sum of spectral intensities for all brain elevated proteins showed no difference between patients and controls (Fig. 3F). In addition to the tissue categories defined by mRNA data, the Human protein atlas gives an expression summary for each protein, this is based on UniProt protein existence; a Human Protein Atlas antibody- or RNA based score and evidence based on PeptideAtlas (The Human Protein Atlas, 2018; Uhlen et al., 2015). Although this summary is not given as a score we have used it to eliminate the proteins with the lowest specificity. Five of the proteins with a high expression in the brain based on the protein expression summary had significant different abundancies in healthy controls compared to patients (Fig. 3, Table 2, Student’s t-test with correction for multiple hypothesis testing by using permutation-based FDR <0.01 and artificial within group variance s0 = 0.1).

Figure 3 Total amount of brain-elevated proteins and significantly changed brain proteins.

Scatter plot for normalized (divided by total sum within each sample) average precursor intensity with mean (bar) and standard deviation (error bars) for the five brain proteins identified as significantly different between psychotic patients during psychosis (T1) and/or in improved state (T2) and healthy controls (HC). (A) Neurogranin (Q92686), (B) Neuron-specific calcium-binding protein hippocalcin (HPCA, P84074), (C) Kalirin (O60229), (D) Beta-adducin (P35612), (E) Ankyrin-2 (Q01484) and (F) all 45 brain-elevated proteins.

Table 2 Significantly changed brain proteins.

Overview of the five brain proteins identified as different between healthy controls (HC) and patients with psychosis (T1) and in improved state (T2). The table shows accession number in Uniprot, molecular weight (MW), number of validated peptides and spectra across all samples, and Average precursor intensity within each group given as mean ± standard deviation (number of samples where the protein is identified).

Uniprot Accession	Protein name	MW (kDa)	#Validated Peptides	#Validated Spectra	Average precursor intensity (Mean ± SD (n))	
					HC	T1	T2	
Q92686	Neurogranin	7.6	2	76	146,020 ± 80,464 (20)	130,421 ± 153,699 (9)	195,775 ± 403,355 (10)	
P84074	Neuron-specific calcium-binding protein hippocalcin	22.4	4	80	188,914 ± 114,734 (14)	281,593 ± 145,068 (21)	263,648 ± 145,611 (16)	
O60229	Kalirin	340.0	11	106	229,014 ± 142,100 (9)	337,939 ± 176,321 (19)	330,904 ± 115,006 (11)	
P35612	Beta-adducin	80.8	13	99	305,652 ± 169,514 (20)	165,875 ± 63,467 (3)	309,466 ± 201,639 (5)	
Q01484	Ankyrin-2	433.4	6	53	295,801 ± 11,1670 (16)	122,702 ± 9,911 (2)	556,426 ± 745,193 (4)	

Membrane-bound protein candidates for immunolabeling of brain-derived EVs

To enable isolation of EVs from the brain, surface proteins with high specificity to the brain can be targeted either in immunoaffinity chromatography or fluorescence activated cell sorting (FACS). We have used the expression summary in the human protein atlas to identify the most brain-specific surface proteins among our identified brain elevated proteins; Plexin B3 (PLXNB3) C type lectin domain 2 family L (CLEC2L), myelin basic protein (MBP) and potassium voltage-gated channel subfamily A member 2 (KCNA2) (The Human Protein Atlas, 2018; Uhlen et al., 2015). We also detected purinergic receptor P2Y12 (P2RY12), a surface protein with high expression in microglia (The Human Protein Atlas, 2018).

Discussion

We present the first characterization of blood-based EVs isolated from psychotic patients with extensive peripheral blood EV proteomes for psychotic patients and healthy persons (Braga-Lagache et al., 2016), contributing to the construction of a comprehensive proteomic database for EV proteins (Choi et al., 2015). Size, concentration and protein concentration in EVs were all increased in psychotic patients compared to controls and remained unaltered with clinical improvement. Use of illegal substances or duration of the psychotic disorder had no influence on EV characteristics, indicating that the findings correlate to other factors than an unhealthy life style which is common in patients with psychosis (Jakobsen et al., 2018). This could suggest that our findings are related to the psychiatric condition itself rather than representing confounders.

Brain-elevated proteins derived from EVs

We identified several brain-specific and brain-elevated proteins in isolated EVs from psychotic patients and healthy controls. Brain-specific proteins have previously been found in EVs from patients with malignant glioma, Alzheimer’s disease (AD), frontotemporal dementia and healthy controls (Galbo Jr et al., 2017; Goetzl et al., 2016a), indicating that EVs originating from the brain can enter the bloodstream. The most likely route is via the brain glymphatic system that can transport large molecules and cells (Louveau et al., 2015). Transport via the blood–brain barrier (BBB) may also contribute this, as preclinical evidence suggests an inflammatory dose-dependent transcytosis of EVs through the BBB (Andras et al., 2017; Matsumoto et al., 2017). Comparing the average content of all brain-elevated proteins, we noticed no difference between patients and controls. Although a crude estimate, this suggest that the difference between patients and controls in terms of total production, clearance by the glymphatics and passage of brain EVs through the BBB is minor.

Neurogranin, neuron-specific calcium-binding protein hippocalcin, kalirin, beta-adducin and ankyrin-2

We found 5 brain proteins with different abundances in patients and controls; all these proteins are involved in the regulation of glutamatergic synapses. GO analysis pointed in the same direction as analysis of individual brain proteins; the GO terms main axon and postsynapse had higher abundance in healthy controls. The formation of neuronal circuits during brain development and their subsequent modification during lifetime require plasticity at excitatory synapses, manifested by changes in synaptic strength (Hanley, 2018). Long-term potentiation (an increase in synaptic strength) and long-term depression (a decrease in synaptic strength) are synapse-specific forms of plasticity (Hanley, 2018). Two important postsynaptic processes are involved in the plasticity of glutamatergic synapses: modifications in the amount of α-amino-3-hydroxy-5-methyl-4-isoxazolepropionic acid receptors (AMPAR) and morphological alterations of dendritic spines mainly mediated by actin filament (Bosch & Hayashi, 2012). Dendritic spines are the postsynaptic structural correlate of excitatory synapses (Han, Cooke & Xu, 2017).

AMPAR is a glutamate receptor that mediates the majority of fast synaptic excitation in the central nervous system. To modulate the synaptic transmission strength, AMPAR are transported to or from synapses (Hanley, 2018). An excitatory synapse containing N-methyl-D-aspartate receptors (NMDAR), but no AMPAR, is termed a silent synapse because of its low activity (Han, Cooke & Xu, 2017). One of the main regulators of AMPAR endocytosis is stimulation of NMDAR (Hanley, 2018).

Hippocalcin, a Ca2+-sensing protein, is found in the retina and the brain and is among the proteins with the highest expression in the brain (Hanley, 2018; The Human Protein Atlas, 2018; Uhlen et al., 2015). Hippocalcin was increased in EVs from our psychotic patients (Fig. 3B). Hippocalcin is required for long-term depression in the synapse, and a suggested mechanism is that hippocalcin recruits AMPAR to endocytic sites in response to NMDAR mediated Ca2+ signals (Hanley, 2018).

Kalirin levels were also increased in EVs from patients (Fig. 3C). Kalirin expression is enriched in the forebrain. Its most abundant isoform, kalirin-7, is localized to dendritic spines on cortical pyramidal neurons, where it plays a key role in morphological and functional plasticity at excitatory synapses and facilitates actin remodeling such that overexpression increases the number of dendritic spines (Penzes & Remmers, 2012). Kalirin-7 interacts with the protein product of, DISC1, modulating the response to NMDAR activation (Hayashi-Takagi et al., 2010; Tropea et al., 2018). When the DISC1 protein are disrupted it predisposes the carrier to a number of mental health disorders including schizophrenia (Sachs et al., 2005; Thomson et al., 2016; Tropea et al., 2018). In our study, the levels of neurogranin, beta-adducin, and ankyrin-2 were lower in EVs from psychotic patients (Figs. 3A, 3D and 3E). The expression of beta-adducin is mainly restricted to the brain and hematopoietic tissues (The Human protein Atlas, 2018; Uhlen et al., 2015) and regulates dendritic spine stability through actin-based synapse formation and spectrin-based synapse stabilization (Engmann et al., 2015). Ankyrin-2 is a member of the ankyrin family of proteins that link the integral membrane proteins to the underlying spectrin-actin cytoskeleton and is a key presynaptic target of casein kinase 2 to maintain synapse stability (Bulat, Rast & Pielage, 2014). Neurogranin, a neuron-specific and postsynaptic protein, increases synaptic strength in an activity- and NMDAR-dependent manner (Zhong et al., 2009). Decreased neurogranin levels lead to accelerated spine elimination and impaired recruitment of AMPAR to silent synapses (Han, Cooke & Xu, 2017).

Our results indicate weakening of the glutamatergic synapse in psychotic patients; the high levels of kalirin being an exception. The role of the detected synapse proteins as well as underlying mechanisms of synaptic plasticity in general, remains little understood (Dieterich & Kreutz, 2016). However, the literature does indicate an important function of EVs in synapse regulation (Ashley et al., 2018; Chivet et al., 2014; Fowler, 2019; Fruhbeis et al., 2013) as is also apparent in our study.

Glutamatergic neurotransmission and psychotic disorders

One of the main hypotheses regarding the pathophysiology of psychotic disorders is abnormal glutamatergic neurotransmission and NMDAR hypofunction (Balu & Coyle, 2015). This is supported by the fact that psychosis typically starts during adolescence, a period involving modification of synapses (Keshavan et al., 2014). Further, NMDAR antagonists can produce psychotic symptoms (Balu & Coyle, 2015; Thiebes et al., 2017). Growing genetic data supports the association between schizophrenia and glutamatergic synapse hypofunction (Fromer et al., 2014; Kirov et al., 2012; Schizophrenia Working Group of the Psychiatric Genomics, 2014). Two large genome wide studies have identified the GO term “abnormal long-term potentiation” on their top list of gene sets enriched in schizophrenia (Pardinas et al., 2018; Pocklington et al., 2015). Two reviews also pointed at variants in genes belonging to the postsynaptic density at the glutamatergic synapse (Hall et al., 2015; Soler et al., 2018). Animal models have provided possible mechanisms linking NMDAR hypofunction to the perceptual disturbances and abnormal associative learning in schizophrenia (Clifton, Thomas & Hall, 2018; Ranson et al., 2019). To summarize, evidence points at dysfunction of the glutamatergic synapse as a possible mechanism in the pathophysiology of psychosis. Our results support data suggesting glutamatergic dysfunction in psychosis and indicate a role of EVs in disease-related synaptic regulation.

Protein candidates for immunolabeling of brain-derived EVs

Our proteomic analysis revealed presence of brain-derived EVs in the blood, in a mixture of EVs from other tissues. The isolation of brain-derived EVs from the blood EV population can enable more detailed analysis on EVs originating directly from the brain. To enable such isolation, affinity methods based on antibodies recognizing surface proteins can be applied. Neural cell adhesion molecule 1(NCAM1), L1CAM and glutamine aspartate transporter (SLC1A3) are proteins that have been applied for immunolabeling of brain-derived EVs (Fiandaca et al., 2015; Goetzl et al., 2015; Goetzl et al., 2016b; Kapogiannis et al., 2015; Kapogiannis et al., 2019b; Mustapic et al., 2017). These proteins were not detected in this study indicating low concentrations. L1CAM and NCAM1 have a low specificity for the brain and are thus not suited for the isolation of pure fractions of brain-derived EVs (The Human Protein Atlas, 2018; Uhlen et al., 2015). Our study identified membrane proteins with high brain specificity (The Human Protein Atlas, 2018). The identified proteins are promising candidates to isolate brain-enriched fractions of EVs, representing different brain cells and compartments. Although purinergic receptor P2Y12 also is highly expressed in peripheral immune cells, this protein is interesting as a target protein for immunolabeling of EVs due to its high expression in microglia. EVs from microglia will be relevant to investigate in the future, as evidence suggests that microglia might contribute to neuroinflammation in psychosis but the usefulness of PET and the translocator protein tracer to assess microglia activation in patients with psychosis has been challenged (Kroken et al., 2018).

GO analysis

GO overrepresentation analysis of significantly changed proteins revealed that immunoglobulin complex, complement pathway and lipoprotein particle-related proteins were overrepresented GO terms for proteins with significantly lower abundance in patients with psychosis. These proteins are large and abundant in human plasma (Braga-Lagache et al., 2016). The apparent increase in these GO pathways could be caused by a proportionally larger co-precipitation of free proteins and a lower overall EV concentration in healthy controls. Cholesterol and lipid-soluble proteins are also present within EVs and the detected apolipoproteins may originate from the EVs themselves (Raposo & Stoorvogel, 2013). Also, there is increasing evidence that EVs carry complement factors as cargo and on their surface, thereby contributing to both pro- and anti-inflammatory immune states (Karasu et al., 2018). Of note, genetic variations in some complement genes and changed levels of complement components are associated with psychosis (Woo et al., 2019).

GO terms overrepresented in proteins significantly higher in patient samples compared with healthy controls were terms related to localization and transportation within and out of the cell, as well as proteins related to activation of neutrophils and other leukocytes. This difference may indicate more active secretion and loading of EVs in psychotic patients. According to two recent meta-analyses, the neutrophil-to-lymphocyte ratio is increased in patients with non-affective psychosis and schizophrenia (Karageorgiou, Milas & Michopoulos, 2019; Mazza et al., 2019), and several studies have shown other types of inflammation and immune alterations in psychotic patients (Karageorgiou, Milas & Michopoulos, 2019; Kroken et al., 2018; Pillinger et al., 2018) in line with our GO analyses. Possible mechanisms could be promotion of inflammation and immune activation by EVs trough their regulatory role or inflammation stimulating EV secretion from immune cells and tissues (Slomka et al., 2018).

Methodological considerations

Our study cohort consisted of acutely admitted patients with a primary diagnosis within the psychosis spectrum. This cohort reflects the real-life setting from an acute and emergency psychiatric treatment facility (Zealberg & Brady, 1999). Although a psychiatric cohort as ours is thus subject to heterogeneity, there are valid arguments against categorizing psychotic disorders into too many different diagnostic entities (Castagnini, Munk-Jorgensen & Bertelsen, 2016; Guloksuz & Van Os, 2018; Pries et al., 2018). Even the validity of the distinction between a primary psychosis with comorbid drug abuse and drug-induced psychosis has been questioned (Caton et al., 2007; Mauri et al., 2017; Wearne & Cornish, 2018; Wilson et al., 2018). From a pragmatic point of view, we therefore conclude that our cohort was suitable for the aim of our study, i.e., investigating EVs in patients in the acute phase of psychosis and after improvement.

Selection of anti-coagulant, concentration steps and freezing and thawing EV samples are all steps that may affect the size or concentration of vesicles in the samples (Barrachina et al., 2019). We aimed to start the EV isolation with fresh samples. However, complete EV isolation and further processing of fresh samples were not feasible in a clinical setting. All samples were centrifuged to remove cells (2,000 g) and subsequently pellet large EVs (at 10,000 g) before EV pellets were frozen prior to further analysis, to ensure that all samples were treated equally. An additional centrifugation step to wash the EV samples after freezing were done to remove the largest aggregates that may have formed due to anti-coagulant activity or freezing and thawing, likely causing a loss of vesicles, as some vesicles have aggregated and were removed.

The use of EDTA as anti-coagulant has been shown to stimulate platelet-derived vesicle formation in some studies (Mullier et al., 2013). Thus, we cannot rule out that the use of EDTA or the analysis of frozen EV samples could affect the EV size and concentration as measured with NTA (Fig. 1). Nevertheless, studies have also shown that both anti-coagulant and freezing had limited effect on EVs isolated from platelet poor plasma (Jamaly et al., 2018). Furthermore, we are comparing groups of samples that have been treated identically, to ensure that results are comparable between sample groups.

Regarding the isolation of EVs, our method is simple, and aims to provide a sample of sufficient EVs for further analysis, with high recovery and low specificity that will contain both small and large EVs, but excluding the smallest EVs, because ultracentrifugation was not applied (Thery et al., 2018). As the yield of EVs from a normal-sized peripheral blood sample is modest, our available sample volumes were not sufficient for dividing the samples into more defined, smaller, fractions, e.g., by a density gradient separation method. However, by comparison with Exocarta and gene ontology, we demonstrated that the obtained samples are highly enriched in EVs, and a number of positive EV markers were identified by proteomics. We also identified apolipoproteins and albumin, which are common non-EV contaminant proteins from blood (Thery et al., 2018), indicating the co-isolation also of soluble proteins or protein aggregates not originating from inside the EVs. Preliminary experiments were also done on selected samples, to isolate smaller vesicles from the supernatant by ultracentrifugation at 110,000 g, to evaluate if this could provide us with additional fractions of smaller EVs in sufficient volumes. However, proteomic analysis of such samples revealed predominately abundant plasma proteins, and none of the expected EV protein markers that were identified in the 10,000 g fraction (data not shown). We conclude that the isolation method used in this study yields samples enriched with important EV protein markers and with sufficient EV amounts for in-depth characterization of individual samples.

Strengths and limitations

Our study has limitations that should be acknowledged. First, this was an explorative study with a small and heterogenous patient cohort. Second, we were unable to control for weight, smoking and metabolic factors that are expected to be unequally distributed in psychotic patients and controls. Third, our study may have been subject to selection bias with the most paranoid and anxious patients declining consent; and finally, 28% of patients with psychosis was lost to follow-up. On the positive side, this is the first study that characterizes peripheral EVs in psychosis, i.e., a novel and promising opportunity to identify biomarkers for a major psychiatric disorder.

Conclusions

Blood-borne EVs differ substantially between patients with psychotic disorders and healthy controls. Also, amounts of several proteins involved in the regulation of plasticity of glutamatergic synapses were altered in the psychotic patients supporting evidence of glutamatergic dysfunction in psychosis and indicate a role of EVs in disease-related synaptic regulation. These should be validated and studied in more detail, to progress the understanding of the role of these proteins in psychosis. Thus, collecting peripheral EVs allows access to brain-originating biological material and may provide novel insights about the underlying processes of psychotic disorders.This study also contribute towards the construction of a comprehensive proteome database for EVs, reporting the first proteomic data for EVs in psychosis, and providing data necessary to further elucidate the biogenesis, cargo and pathophysiological role of EVs. We suggest that future studies investigate more thoroughly if potential confounders as lifestyle factors, medications or high stress levels contribute to the changed EV profile in patients with psychotic disorders. The suitability of identified surface brain proteins as tools to isolate a “liquid brain biopsy” should also be evaluated further. Eventually, if our findings on the glutamatergic proteins are confirmed it will be highly relevant to study their mechanistic role as related to EVs.

Supplemental Information

Table S1 List of all identified proteins with accession numbers

List of all identified proteins with accession numbers based on Swissprot database of Homo sapiens (reference proteome downloadedfromUniProtKB [PMID 14681372]in March 2018) with with porcine trypsin (P00761) added as likely contaminant (40660 entries in concatenated database, based on 20330 entries from uniprot.org). In case of protein inference, alternative proteins are given in column ”Other proteins” with Protein Inference Class given. Protein sequence coverage (%) is given. In addition, number of distinct peptides assigned to each protein, as well as spectra and average spectral intensity given for each protein identification in all separate samples are given.

Click here for additional data file.

Table S2A Over-represented GO terms

Over-represented GO terms in the presently identified EV-proteome (all proteins identified in all samples combined) compared to the entire human proteome for Biological process and Cellular compartment GO categories. The table provide number of proteins defined in the GO-term for the reference proteome and our EV proteome, respectively, the expected number of proteins for the EV proteome based on relative representation in the reference proteome and size of the EV-proteome, the difference from the expected number given as fold-enrichment, and the p-value from the Fisher’s exact test and FDR-adjusted values for the fold enrichment.

Click here for additional data file.

Table S2B GO terms for the proteins significantly increased in patients

Over-represented GO terms for the proteins significantly increased in T1 and/or T2 compared with HC, compared to the entire human proteome for Biological process and Cellular compartment GO categories. The table provide number of proteins defined in the GO-term for the reference proteome and our EV proteome (Sample_Up), respectively, the expected number of proteins for the EV proteome based on relative representation in the reference proteome and size of the EV-proteome, the difference from the expected number given as fold-enrichment, and the p-value from the Fisher’s exact test and FDR-adjusted values for the fold enrichment.

Click here for additional data file.

Table S2C GO terms for the proteins significantly decreased in patients

Over-represented GO terms for the proteins significantly decreased in T1 and/or T2 compared with HC, compared to the entire human proteome for Biological process and Cellular compartment GO categories. The table provide number of proteins defined in the GO-term for the reference proteome and our EV proteome (Sample_Down), respectively, the expected number of proteins for the EV proteome based on relative representation in the reference proteome and size of the EV-proteome, the difference from the expected number given as fold-enrichment, and the p-value from the Fisher’s exact test and FDR-adjusted values for the fold enrichment.

Click here for additional data file.

Table S3 Proteins differentially expressed in patients compared with healthy controls

List of proteins differentially expressed in T1 and/or T2 compared with HC. Values given for statistical analysis in Perseus, using Student’s t-test with correction for multiple hypothesis testing by using permutation-based FDR < 0.01 and artificial within group variance s0=0.1. Missing values were imputed from a normal distribution with a 1.8 standard deviation shift from the average and a width of 0.3. Values given are -log(p-value) and fold change (difference in log-values of group average) given for T1 vs HC and T2 vs HC, respectively.

Click here for additional data file.

Table S4 The GO terms “main axon” and “postsynapse”

Proteins classified in the enriched GO terms “main axon” and “postsynapse”, in the group of proteins downregulated in T1 and/or T2 compared with HC (Table S1C).

Click here for additional data file.

Table S5 List of identified apolipoproteins in the EV proteome, with normalized average precursor intensities given as average of each patient group

Click here for additional data file.

Table S6 List of identified proteins in the EV proteome that overlap with the Brain protein atlas protein list downloaded from Human protein atlas

Click here for additional data file.

Data S1 Raw data of particle concentration, particle diameter, protein concentraion, psychotic disorder, substance use and time since debut of first psychosis

Click here for additional data file.

We thank Olav Mjaavatten and Even Birkeland at the Proteomics Unit of the University of Bergen (PROBE) for performing mass spectrometry analysis and Harald Barsnes from the University of Bergen for assistance with proteomic software. We also thank the laboratory and Kjetil Sørensen at Østmarka, St Olavs University Hospital and the Clinic of Laboratory Medicine at St Olavs University Hospital for practical support with blood sampling and initial preparation of samples.

Additional Information and Declarations

Competing Interests

Author Contributions

Human Ethics

Data Availability

Hanne Haslene-Hox and Einar Sulheim are employed as Research Scientists at SINTEF, a non-profit research organization.

The authors declare there are no competing interests.

Mette Elise Tunset conceived and designed the experiments, analyzed the data, prepared figures and/or tables, authored or reviewed drafts of the paper, and approved the final draft.

Hanne Haslene-Hox conceived and designed the experiments, performed the experiments, analyzed the data, prepared figures and/or tables, authored or reviewed drafts of the paper, and approved the final draft.

Tim Van Den Bossche and Daniel Kondziella analyzed the data, authored or reviewed drafts of the paper, and approved the final draft.

Arne Einar Vaaler conceived and designed the experiments, authored or reviewed drafts of the paper, and approved the final draft.

Einar Sulheim performed the experiments, analyzed the data, prepared figures and/or tables, authored or reviewed drafts of the paper, and approved the final draft.

The following information was supplied relating to ethical approvals (i.e., approving body and any reference numbers):

The study was approved by the Regional Ethics Committee, South East Norway (2016/949).

The following information was supplied regarding data availability:

The mass spectrometry data and the identification results are available at ProteomeXchange Consortium via the PRIDE partner repository: PXD016293. Raw data regarding EV and participant characteristics are available as a Supplementary File.

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
