# Peer review of "Extracellular vesicles in patients in the acute phase of psychosis and after clinical improvement: an explorative study"

_PeerJ, doi:10.7717/peerj.9714_

## Round 0.1 · original submission · Major Revisions

Your manuscript has now been seen by 2 reviewers. You will see from their comments below that while they find your work of interest, some major points are raised. We are interested in the possibility of publishing your study, but would like to consider your response to these concerns in the form of a revised manuscript before we make a final decision on publication. We therefore invite you to revise and resubmit your manuscript, taking into account the points raised. Please track all changes in a copy of the manuscript text file.

·

Basic reporting

Here, in this paper, authors found five biomarkers in peripheral blood-derived extracellular vesicles (EVs), which can be used to diagnose psychotic patients. In addition, they found physio-chemical differences of EVs between healthy controls and patients. These findings will impact on further development of diagnostic tool. I think this paper can be accepted with minor revisions. My comments follow in the sections below.

Page numbers were not provided, even though line numbers were, so I will write main text’s section name. For detailed comments like typos, I strongly recommend to provide page number next time. Because the PDF file I got have three additional pages at front added by the editor, I cannot be sure that the page number in Adobe Acrobat will be the same page number that authors see in their own manuscript.

Experimental design

1. Were brain related proteins significantly enriched in this EV proteome? 45 out of 1853 seems to be not that significant and I cannot find how you defined “brain-related” with specific cutoff of Human Protein Atlas. Please notify how you defined “brain-related” and if it is not statistically significant, please note this clearly on the main text.

2. Washing step between LC-MS/MS sample runs can be specified in “Proteomics of isolated EVs” section of “Materials and Methods”.

Validity of the findings

1. PXD016293 in PRIDE database cannot be searched or viewed, because authors kept it private. For revision process, they have to share reviewer ID and password so that reviewers can check their data quality.

Additional comments

1. Instead of UniProt accession, it will be better to provide gene symbol in the main text for smooth reading.

2. References should be formatted properly. For example, some paper titles use capital letter for each word not others use it for the first letter of whole title. Please check page number format (117-122 vs 105-11 / adding p. even though it is not page number but article number, etc) and journal name format (adding period if you use abbreviation, etc).

3. Figure 1 is too wordy and not essential for this paper. This figure seems to be “graphical abstract” that authors might have used for submission to other journals. Authors can summarized it to one figure panel if they want to keep this figure.

4. Figure 2B – For each patient, did you use the exactly the same number of EVs to test size? If not, it will be fair comparison if you change y axis to “relative percentage” not the actual number of EVs. If you use the exactly the same numbers for each sample, please write it down in main text or figure legend.

5. Figure 2A 2C 2D and 4 – I guess, you can see statistical significance because you used “paired” t-test. Please provide the information about sample pairing or use paired ratio (patient/control) to visualize your data more significantly in the figure. It seems not that significant graphically in the figures.

Reviewer 2 ·

Basic reporting

This manuscript is relevant for initiating studies of extraceular vesicles in patients with psychosis . The study group is small, however through the experimental methodology of analyzing proteomic EVs , at least the study allows two interesting objectives such as seeing proteins associated with synaptic regulation and also seeing if there are possible markers in brain-derived EVs circulating in periphery blood.
The greatest weakness of the manuscript is the way of working with samples and with the isolation of extracellular vesicles. The authors do not follow the recommendations given by the ISEV international society of extracellular vesicles that regulates most works in this area.

Experimental design

The sample collect should be made with citrate and not with EDTA a calcium chelating necessary for the production of EVS. The sample should be worked immediately for purifications of EVs and maintained a 4 C and not a -80 C
Unfortunately the comparisons of size , amount of protein and kind of EVs are wrong because to keep at -80 C generate aggregates and break of vesicles. The recomendation from ISEV include more steps and the centrigugations to
eliminate debris and to obtain small and large EVs .

Validity of the findings

My recommendation is to consider total extracellular vesicles and work proteomically with the 45 unique brain molecules and with the putative markers of Vesicules of brain detected in EVS for bioinformatics analysis and with a greater discussion of the relevance of markers and as they could be validated through an experimental design
Re write the manuscript , removing the EVs analisis and to concentrate in proteomic findings .

Additional comments

Due to the current situation of the pandemic and being a study with patients, the authors should make a effort to rewrite with a better rational, shorten the manuscript and dedicate efforts to highlight the importance of the molecules obtained and how it could be validated in the future

Annotated reviews are not available for download in order to protect the identity of reviewers who chose to remain anonymous.

---

## Round 0.2 · accepted · Accept

Thank you for the detailed response letter. We are delighted to accept your manuscript for publication.

·

Basic reporting

No further comment for this secondary revision.

Experimental design

No further comment for this secondary revision.

Validity of the findings

No further comment for this secondary revision.

Additional comments

The authors carefully revised their manuscript, including omission of Fig. 1, which strengthens the manuscript. I want to congratulate the authors to their work and suggest to go ahead with the manuscript.